# RouterInterp: Superposed Specialisation in MoE Routing

**Ilya Lasy**
Faculty of Informatics,
TU Wien, Vienna, Austria
`ilya.lasy@tuwien.ac.at`

**Yinuo Nora Cai**
Independent
`nora.cai@gmail.com`

**Kola Ayonrinde**
UK AI Security Institute, London, UK
`koayon@gmail.com`

## Abstract

Sparse Mixture of Experts (MoE) models scale more efficiently than dense models by routing tokens to modular expert networks that are only active when relevant to the task. A leading hypothesis for the performance of MoE models is that each expert specialises in a single, coherent domain. However, interpretability efforts that assume this hypothesis have generally been unsuccessful. We propose and present evidence for an alternative account that we call the *Superposed Specialisation Hypothesis* (SSH): experts specialise in a disjoint union of fine-grained features rather than one broad domain. Leveraging the SSH, we introduce *RouterInterp*, a method for interpreting expert routing that identifies Sparse Autoencoder features most predictive of routing decisions and produces unified natural language explanations. On gpt-oss-20b, explanations from RouterInterp predict expert routing with 77% higher accuracy than prior methods. This work provides a scalable method for generating concise and more accurate explanations of expert routing and increases our understanding of a previously uninterpretable component of foundation models. [1]

## 1 Introduction

Sparse Mixture-of-Experts (MoE) transformers have emerged as a promising approach for scaling frontier language models (Cai et al., 2025; Fedus et al., 2022b; Du et al., 2022). Unlike dense transformers where each input is processed by all parameters, MoE models contain multiple expert networks and a routing mechanism that selects a subset of these "experts" for each input token at each layer. Sparse MoE models can activate only 2–15% of parameters per input (Shazeer et al., 2017a; Liu et al., 2024), enabling dramatic parameter scaling with minimal increases in inference cost (Fedus et al., 2022a).

The strong performance of Sparse MoE models has often been attributed to expert specialisation (Lewis et al., 2021): if each expert learns to handle a subset of the input data distribution or perform only a subset of computations, then using only a subset of the parameters for each input can be both *effective* and *efficient* We call this explanation for the success of MoEs the **Specialisation Hypothesis**.

One seemingly natural corollary of the Specialisation Hypothesis is that if each expert specialises in a particular domain then routing decisions should be human-interpretable. For example, perhaps one expert specialises in medical text, another in mathematics, and yet another in storytelling (Ayonrinde, 2023a). However, prior work has struggled to recover clear, interpretable patterns of expert specialisation (Jiang et al., 2024; Lewis et al., 2021; Zoph et al., 2022). We believe that this difficulty derives from not distinguishing between two

---

[1] Code available at `https://github.com/koayon/moe-interp`

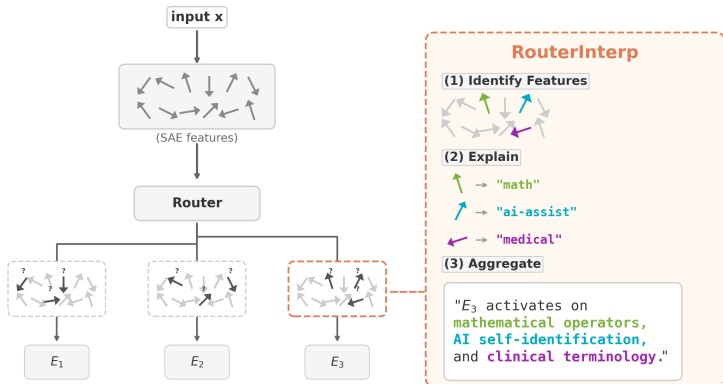

Figure 1: RouterInterp explains routing decisions as a combination of interpretable features. Sparse autoencoder (SAE) latents, sparse feature representations extracted from model activations, pick out directions in activation space that correspond to interpretable features. RouterInterp *identifies* SAE latents that are most predictive of routing, *explains* these features in concise natural language strings, and *aggregates* these explanations into an expert-level explanation. While no single monosemantic concept captures an expert's behavior, aggregating multiple features makes routing understandable.

distinct forms of the Specialisation Hypothesis which we term the **Domain Specialisation Hypothesis** (DSH) and the **Superposed Specialisation Hypothesis** (SSH), respectively. The distinction between these hypotheses becomes apparent when we consider that real-world data contains far more fine-grained categories (e.g., specific topics, syntactic constructions, reasoning patterns) than any MoE layer has experts.[2] This means multiple such categories, which we call *micro-domains*,[3] must inevitably be routed to the same expert [4]. Under the *Domain Specialisation Hypothesis*, semantically similar micro-domains cluster within an expert, yielding a coherent domain that the expert specialises in. In contrast, under the *Superposed Specialisation Hypothesis*, an expert specialises in a disjoint collection of features spanning multiple unrelated micro-domains. This mirrors the *superposition* phenomenon observed in neural networks, where multiple features are represented in the same set of neurons (Elhage et al., 2022).

> **Specialisation Hypotheses**
>
> **Domain Specialisation Hypothesis:** Sparse Mixture-of-Experts models work well because different experts specialise in different semantically coherent domains (e.g. medical text, mathematical reasoning, romance language translation).
>
> **Superposed Specialisation Hypothesis:** Sparse Mixture-of-Experts models work well because different experts specialise in disjoint collections of features corresponding to multiple micro-domains that are not highly semantically related.

Our work tests these two hypotheses by analysing expert routing decisions in large MoE models using methods inspired by each hypothesis. We find evidence for the Superposed Specialisation Hypothesis (SSH): our experiments show that expert routing can be accurately predicted by a combination of micro-domain features but not by a single monosemantic domain (Section 5). We also present theoretical arguments for the SSH (Section 3). Based on these findings, we develop **RouterInterp**, a new method that achieves state of the art performance in producing accurate and concise explanations of MoE routing decisions.

---

[2]Typical MoE models use between 8 and 128 experts per layer (Jiang et al., 2024; Lewis et al., 2021; Zoph et al., 2022).

[3]For example, "medical text" may be a broad domain composed of micro-domains such as "anatomy", "diseases", and "medical procedures".

[4]by the Pigeonhole Principle (Rebman, 1979)

While previous router interpretation methods focused on per-token routing statistics, *Router-Interp* instead interprets routing in terms of contextual features extracted from model activations via sparse autoencoders (SAEs) (Makhzani & Frey, 2013; Cunningham et al., 2024; Bricken et al., 2023). SAEs decompose activations into sparse latent representations (SAE latents), where each latent corresponds to an interpretable feature (Section 2.1). Inspired by the Superposed Specialisation Hypothesis, we interpret routing decisions as arising from a *combination* of features. We identify which of the SAE latents are most predictive of each expert's routing behaviour by measuring how much more strongly each feature activates when a particular expert is selected (Section 4.2).

To evaluate whether *RouterInterp*'s explanations are human-understandable, we measure how accurately a language model can predict expert activation given only the explanation (Section 4.4). *RouterInterp* achieves 89% and 81% accuracy in predicting expert routing based on SAE features on OLMoE-1B-7B and gpt-oss-20b, respectively. RouterInterp attains a 0.62 explanation score on gpt-oss-20b, outperforming explanations based on token-expert co-occurrence (0.35). By interpreting routing as a superposition of interpretable SAE features rather than a single domain, RouterInterp demonstrates that we can meaningfully understand MoE routing decisions even when experts specialise in disjoint collections of features.

Our contributions are as follows:

- We provide two theoretical arguments for the Superposed Specialisation Hypothesis: first, reducing interference between in-expert computations, and second, improving the efficiency of expert utilisation under the auxiliary load-balancing losses typical in MoE training (Section 3).
- We present *RouterInterp*, our SAE-based method, which produces accurate and concise explanations of routing decisions, achieving 0.62 explanation score and 81% routing prediction accuracy on gpt-oss-20b (Section 4).
- We provide experiments to suggest that expert specialisation is best understood through the Superposed Specialisation Hypothesis: the idea that experts specialise in a disjoint rather than semantically coherent collection of features. This implies that MoE routing decisions can be understood as mediated by a combination of features rather than a single monosemantic concept or "domain" of specialisation (Section 5).

## 2 BACKGROUND

### 2.1 SPARSE AUTOENCODERS (SAE)

A fundamental challenge in interpreting LLM activations is *superposition*: models encode more features than the number of available dimensions, resulting in *polysemantic neurons* that activate for multiple unrelated concepts (Elhage et al., 2022). Sparse Autoencoders (SAEs) address this by mapping an activation vector $\boldsymbol{x} \in \mathbb{R}^N$ to a corresponding sparse latent representation $\boldsymbol{z} \in \mathbb{R}^F$.[5] Ideally, $\boldsymbol{z}$ should represent an "unpacking" of the compressed representations into monosemantic, single-concept latents referred to as *features* (Bricken et al., 2023; Cunningham et al., 2024).

The SAE architecture consists of an encoder that projects activations into a sparse latent space, and a decoder that reconstructs the original activation from the sparse latents:

$$\boldsymbol{z} = \sigma_s(\boldsymbol{W}_{\text{enc}}(\boldsymbol{x} - \boldsymbol{b}_{\text{pre}}) + \boldsymbol{b}_{\text{enc}}) \tag{1}$$

$$\hat{\boldsymbol{x}} = \boldsymbol{W}_{\text{dec}}\boldsymbol{z} + \boldsymbol{b}_{\text{pre}} \tag{2}$$

where $\boldsymbol{W}_{\text{enc}} \in \mathbb{R}^{N \times F}$, $\boldsymbol{W}_{\text{dec}} \in \mathbb{R}^{F \times N}$, and $\sigma_s$ is a sparsifying activation function. We follow Gao et al. (2025)'s **Top-K SAE** approach, where $\sigma_s = \text{TopK}(\cdot, k)$ retains only the $k$ largest activations per input and zeros out the rest, ensuring that the latent is sparse. The SAE is trained to optimise reconstruction fidelity $\mathcal{L} = \|\boldsymbol{x} - \hat{\boldsymbol{x}}\|_2^2$, ensuring that the original activation can be accurately reconstructed from the sparse latent representation $\boldsymbol{z}$.

---

[5]Typically $F > N$, allowing the SAE to represent more features than the dimensionality of the original activation vector $\boldsymbol{x}$.

## 2.2 Mixture of Experts (MoE)

In a standard MoE layer, the dense feed-forward network (FFN) is replaced by $E$ parallel expert networks $\{\boldsymbol{E}_i\}_{i=1}^E$, each typically an FFN itself. A learned router network ($\boldsymbol{W}_r$) determines which experts process each token. Given an input token representation $\boldsymbol{x}$, the router computes routing logits $h(\boldsymbol{x}) = \boldsymbol{W}_r \cdot \boldsymbol{x}$ and converts them to a probability distribution over experts via softmax:

$$p_i(\boldsymbol{x}) = \frac{\exp(h(\boldsymbol{x})_i)}{\sum_{j=1}^E \exp(h(\boldsymbol{x})_j)} \tag{3}$$

To enforce sparsity, only the top-$k$ experts with the highest probabilities are selected. Letting $\mathcal{T}$ denote the set of selected expert indices, the layer output is computed as the weighted sum of expert outputs:

$$\boldsymbol{y} = \sum_{i \in \mathcal{T}} p_i(\boldsymbol{x}) \cdot \boldsymbol{E}_i(\boldsymbol{x}) \tag{4}$$

A key challenge in MoE training is *load balancing*: without intervention, models tend to collapse to using only a few experts, leaving others undertrained (Shazeer et al., 2017b). This is addressed through auxiliary losses that encourage uniform expert utilization.

## 3 Specialisation Hypotheses

In Section 1, we introduced two ways to operationalise the idea of MoE models being effective due to specialisation in terms of the *Domain Specialisation Hypothesis (DSH)* and the *Superposed Specialisation Hypothesis (SSH)*. These hypotheses can be understood as claims about whether expert routing is *monosemantic* (can be explained by a single concept or domain) or *polysemantic* (can only be explained by a collection of multiple concepts or domains which are relevant in different contexts). The idea of superposition is inspired by Elhage et al. (2022), who describe a theory of the superposition of neurons.[6] We extend this idea to the superposition of *experts*, arguing that each expert may be specialized in a disjoint collection of domains.

In this section, we formalise the two hypotheses (DSH and SSH), state their predictions, and give theoretical motivations for why SSH might hold in practice.

### 3.1 Alternative Specialisation Hypotheses

Suppose that we have a set of $D$ micro-domains occurring in a corpus $\mathcal{C}$, denoted $\mathcal{D} = \{d_1, \ldots, d_D\}$.[7] Suppose also that we have an MoE layer with $E$ experts.

Firstly, note that when $D = E$, that is there are the same number of micro-domains as experts, then we should expect the experts to specialise in a single micro-domain. This is consistent with both the DSH and SSH. Secondly, when $D < E$, that is there are fewer micro-domains than experts, then we should expect multiple experts to specialise in the same micro-domain or for there to be redundant experts which are never routed to. This is also consistent with both the DSH and SSH.

However, in the more interesting and realistic case of $D > E$, the two hypotheses make different predictions. Under the DSH, similar micro-domains are routed to the same expert and so an expert specialises in a semantically coherent domain consisting of adjacent micro-domains - expert specialisation can be explained by a single (monosemantic) concept or domain. Under the SSH, dissimilar micro-domains are routed to the same expert and so an expert specialises in a collection of semantically disjoint domains - expert specialisation can only be explained by multiple (polysemantic) concepts or domai

---

[6]Arora et al. (2018) and Goh (2016) both find evidence for superposition-like phenomena and utilise this structure for interpretability similarly to our work.

[7]Here, the domains could be topics, genres, or other semantic or syntactic categories in the data or could alternatively represent sections of the input space that require the same processing for the next layer of the model.

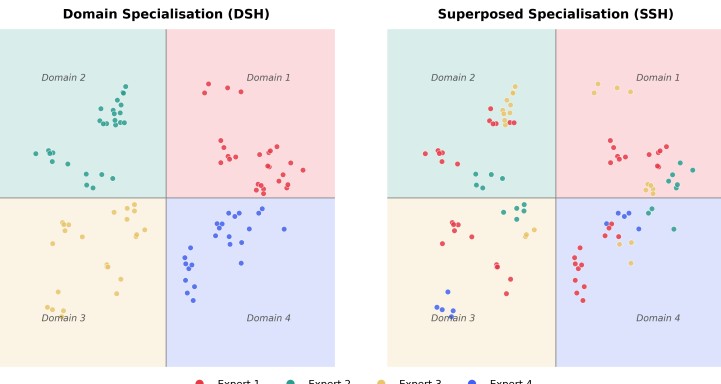

Figure 2: Illustration of two alternative hypotheses for the expert specialisation in MoE models, (a) the Domain Specialisation Hypothesis (DSH) and (b) the Superposed Specialisation Hypothesis (SSH). (a) Under the DSH, dataset examples (points) from the same semantic domain (coloured regions) are routed to the same expert (they all have the same colour). In this case experts specialise in a single coherent domain. (b) Under the SSH, however, examples routed to the same expert (i.e. have the same colour) are scattered across multiple disjoint domains (different coloured regions). Here the clusters of adjacent points that are routed to the same expert represent *micro-domains*: fine-grained features of the input dataset. In this case, experts do not specialise in a single domain but instead specialise in a disjoint union of fine-grained features (micro-domains). We suggest that the SSH is a more accurate description of expert specialisation in practice.

## 3.2 THEORETICAL MOTIVATION FOR THE SSH

There are two core arguments for why we might expect the Superposed Specialisation Hypothesis to be a more accurate description of expert specialisation.

**Argument 1 - Interference Minimisation.** Elhage et al. (2022) argue that superposition is an effective strategy for models because two sparse features that are not frequently co-activated can relatively unambiguously share the same set of neurons. Sparsity, and in particular, low co-activation, reduces the interference cost of superposition and so allows the model to represent more features than the number of available neurons. They find that correlated features tend to be represented orthogonally because otherwise this would lead to high levels of interference noise that would make it difficult for the model to disentangle and use these features.

Analogously, we argue that the router is incentivised to assign dissimilar domains to the same expert so that the expert can perform Computation in Superposition (Hänni et al., 2024; Linsefors & Bushnaq, 2025a;b; Newgas, 2025). In other words, a single expert can have multiple disjoint transforms that it applies depending on the input domain.

**Argument 2 - Load Balancing.** In MoE training, it is common to include a load-balancing loss that encourages the expert router to distribute the incoming tokens evenly across all experts to avoid the under-utilisation of some experts (Shazeer et al., 2017a; Fedus et al., 2022a). If models are additionally trained with relatively small batch sizes, then this can have the unintended effect of encouraging the expert router to route dissimilar inputs to the same expert. We can see this because a single sequence is likely to contain mainly tokens that are from the same macro-domain. Hence, if the batch size is small, then the batch will not contain very many different macro-domains; however, the load-balancing loss will encourage the router to distribute tokens from these few macro-domains across all of the experts. This necessarily requires that some tokens from the same macro-domain will be routed to different experts - in opposition to the DSH, where we would expect all tokens from the same macro-domain to be routed to the same expert.

# 4 METHODS

If the SSH is true, then we should expect that each expert can be activated by a range of disparate features and hence that it is generally not possible to explain an expert's routing behaviour in terms of a single coherent domain. We operationalise a test of this hypothesis as follows. First, we train classifiers to predict expert routing from SAE feature activations, where the features are proxies for micro-domains (Section 4.1). We then present **RouterInterp**, our method for generating natural language explanations of expert routing through aggregating feature explanations (Section 4.2).

## 4.1 SAE-BASED ROUTING PREDICTION

Before interpreting routing via SAE features, we first evaluate whether SAE features can linearly predict routing patterns. We extract activations $x$ from the residual stream before the routing function and use an SAE encoder to produce sparse feature representations $z$. We then train a logistic regression classifier on these sparse feature representations, one per expert $E_i$, to predict whether a given expert is activated for some context.

To investigate how routing information is distributed across features, we additionally train classifiers using only the $m$ most active features per token (ranked by activation magnitude). By varying $m$, we can assess how polysemantic expert routing is: if many features are required to predict routing decisions with high accuracy, this indicates that multiple concepts are routed to the same expert in superposition.

## 4.2 ROUTERINTERP

RouterInterp operates in three stages: (1) identifying SAE features most relevant to each expert's routing behaviour, (2) generating natural language explanations for each feature, and (3) aggregating these into expert-level explanations.

**Feature Selection.** We identify which SAE features are most predictive of each expert's activation using the '$\rho$-usefulness' metric (Ilyas et al., 2019), a data-driven selection criterion that measures how discriminative a feature is for expert selection. A feature $f$ is $\rho$-useful for expert $E_i$ if the expected activation of $f$ is higher when expert $E_i$ is selected than when it is not:

$$\rho(f, E_i) = \mathbb{E}[f(\boldsymbol{x}) \mid E_i \in \mathcal{T}(\boldsymbol{x})] - \mathbb{E}[f(\boldsymbol{x}) \mid E_i \notin \mathcal{T}(\boldsymbol{x})] \tag{5}$$

where $\mathcal{T}(\boldsymbol{x})$ denotes the set of selected experts for input $\boldsymbol{x}$. Features with high $\rho$-usefulness scores are specifically predictive of a particular expert's activation. For each expert $E_i$, we select the top-$n$ features ranked by $\rho$-usefulness score, denoting this set of features $\mathbb{F}_i$. [8]

**Feature Explanations.** Given the selected feature set $\mathbb{F}_i$ for each expert $E_i$, we generate natural language explanations for each feature following the automated interpretability framework of (Paulo et al., 2025). For each feature $f \in \mathbb{F}_i$, we present a language model (the *explainer*) with *positive examples* where $f$ activates strongly and *contrastive negatives* where $f$ does not activate despite being semantically similar (in embedding space). The explainer infers what concept or pattern the feature detects, producing a concise natural language description.

**Aggregation.** To obtain an expert-level explanation, we prompt a language model with all $n$ feature explanations for expert $E_i$ and ask it to synthesise a single coherent paragraph describing when the expert activates. The goal is an explanation that enables predicting when the expert activates; we evaluate this using an LLM as a proxy for human judgment (Section 4.4).

---

[8]We evaluate the effect of feature set size $|\mathbb{F}_i|$ on explanation quality and compare against an alternative selection method based on cosine similarity between router weight vectors and SAE decoder directions in Section D.

### 4.3 N-gram Baseline

Prior work on MoE interpretability has relied on token-level analysis to understand expert specialisation, with mixed results: Jiang et al. (2024) found no obvious patterns in expert assignment, while Tigges (2025) reported limited success for specific experts. We formalise and extend their token-level analysis as a baseline for comparison.

We use n-gram statistics in two ways. First, as a **routing prediction baseline**: for each token, we count how often it co-occurs with each expert's activation on a training set and predict that a token routes to its most frequent experts. This directly compares to our SAE-based predictor (Section 4.1). Second, as an **explanation baseline**: for each expert, we prompt a language model to summarise the most frequently co-occurring tokens into a natural language description, evaluated using the same scoring procedure as RouterInterp (Section 4.4). We extend prior unigram-only analyses to also include bigrams.

### 4.4 Evaluation

**Routing Prediction.** To evaluate the accuracy of our SAE-based routing prediction, we report *Recall* as our main evaluation metric, measuring the proportion of experts that are correctly predicted as being routed to. [9] For a token $t$ with actual expert set $E_{\text{actual}}(t)$ and predicted expert set $E_{\text{pred}}(t)$, both of size $k$:

$$\text{Recall} = \frac{1}{N} \sum_{t=1}^{N} \frac{|E_{\text{pred}}(t) \cap E_{\text{actual}}(t)|}{|E_{\text{actual}}(t)|} \tag{6}$$

where $N$ is the total number of tokens.

**Explanation Scoring.** To evaluate whether our expert explanations accurately capture routing behaviour, we adapt the AutoInterp scoring framework from Paulo et al. (2025). In AutoInterp, another language model (the *scorer*) uses the explanation to predict expert activation on held-out examples.

We construct a balanced evaluation set for each expert $E_i$ containing *positive examples* (contexts where a token was routed to expert $E_i$) and *hard negatives* (semantically similar contexts without tokens routing to $E_i$). To find hard negatives, we embed all contexts using a sentence embedding model, rank by cosine similarity to each positive, and select the most similar contexts that: (1) contain no tokens routing to $E_i$, and (2) contain tokens routing to some expert $E_j$ sharing features with $E_i$ ($\mathbb{F}_i \cap \mathbb{F}_j \neq \varnothing$). For each example in the evaluation set, we present the scorer with the example's textual context and the expert explanation, asking it to predict whether expert $E_i$ would be activated. This yields a binary prediction per example, which we compare against the ground truth routing decision. We report F1 score of this binary classification task, measuring how well the synthesised explanation captures the expert's routing behaviour.

## 5 Results

### 5.1 Routing Prediction

To show that token-based statistics can't adequately predict expert routing, we compare n-gram baselines against expert prediction based on SAE latents.

We evaluate whether SAE latents can predict MoE routing decisions across different model architectures. Table 1 compares our SAE-based predictor against n-gram baselines on both OLMoE-1B-7B and gpt-oss-20b. The SAE predictor achieves 0.89 and 0.81 Recall respectively, substantially outperforming both baselines across both architectures. N-gram baselines perform markedly worse on the larger model (0.74 for OLMoE vs 0.55 for gpt-oss),

---

[9]Since both the predicted set $E_{\text{pred}}(t)$ and actual set $E_{\text{actual}}(t)$ have cardinality $k$ by construction (due to top-$k$ routing), Recall equals Precision and F1 in this setting (equal set sizes imply FP = $k - |E_{\text{pred}} \cap E_{\text{actual}}|$ = FN).

|                       | OLMoE-1B-7B | gpt-oss-20b |
|-----------------------|:-----------:|:-----------:|
| Unigram Baseline      | 0.70        | 0.52        |
| Bigram Baseline       | 0.74        | 0.55        |
| SAE Predictor (Ours)  | **0.89**    | **0.81**    |

Table 1: Token-level statistics are insufficient to predict MoE routing: richer representations are required. N-gram baselines, which predict routing from token co-occurrence frequencies, achieve only 0.52–0.74 Recall, while our SAE-based predictor achieves 0.81–0.89 by leveraging learned feature representations. The performance gap widens on the larger model (gpt-oss-20b), suggesting routing becomes increasingly context-dependent at scale. This motivates using SAE features, rather than tokens, as the basis for interpreting expert routing. Results reported as Recall for layer 11 of OLMoE-1B-7B and layer 16 of gpt-oss-20b.

|                       | Layer 4  | Layer 8  | Layer 12 | Layer 16 | Layer 20 |
|-----------------------|:--------:|:--------:|:--------:|:--------:|:--------:|
| Unigram Baseline      | 0.57     | 0.60     | 0.52     | 0.57     | 0.55     |
| Bigram Baseline       | 0.63     | 0.64     | 0.56     | 0.59     | 0.59     |
| SAE Predictor (Ours)  | **0.82** | **0.85** | **0.79** | **0.82** | **0.83** |

Table 2: The advantage of SAE-based prediction over token co-occurrence frequencies holds across all network depths. SAE-based prediction achieves 0.79–0.85 Recall across layers 4–20 of gpt-oss-20b, while n-gram baselines plateau at 0.52–0.64. The narrow range of SAE predictor performance suggests routing complexity is similar across layers.

suggesting that routing decisions in larger models are more context-dependent and cannot be predicted from surface-level token statistics alone. In contrast, the SAE predictor maintains strong performance across both architectures.

Table 2 shows that the advantage of SAE-based prediction holds across all layers of gpt-oss-20b. The consistent advantage at both early and late layers indicates that routing is better explained by SAE latents than by token statistics, supporting our hypothesis that routing operates at the feature level. We provide a detailed ablation on the number of active SAE latents $m$ needed to outperform n-gram baselines in Section C.

## 5.2 Evidence for Superposed Specialisation

The preceding results show that SAE features predict routing better than tokens, but this alone does not distinguish the DSH from the SSH: multiple features could still represent semantically related concepts within a single coherent domain. We present two experiments that directly test whether expert-predictive features represent disjoint micro-domains (full details in Section E).

**Feature co-activation analysis.** For each expert, we measure pairwise co-activation rates among its top-20 $\rho$-useful features over ~10M tokens. Under the DSH, features within the same expert should represent related concepts and co-activate well above the rate expected under statistical independence; under the SSH, they should co-activate at approximately the independence baseline. We normalise observed co-activation by the independence baseline $\bar{p}^2$ (where $\bar{p}$ is the mean firing rate), yielding a ratio that equals 1.0 when features fire independently. Across 128 experts (32 per layer) in gpt-oss-20b, observed co-activation closely follows the $\bar{p}^2$ independence curve rather than the $\bar{p}$ perfect-correlation line (Figure 4). These results indicate that expert-predictive features represent semantically disjoint micro-domains rather than facets of a single coherent domain.

**The Pile case study.** We additionally verify that experts are not domain-specialised at the macro level by measuring the distribution of Pile dataset (Gao et al., 2020) subsets routed to each expert. We compute the normalised entropy of each expert's subset distribution, where 1.0 corresponds to uniform routing and lower values indicate domain preference. Across all layers, expert entropies cluster near the corpus-proportional baseline (median entropy 0.74; a non-specialised expert mirroring the global corpus mix would score 0.80), indicating that experts process a broad mix of domains rather than specialising in any single one (Figure 5).

|              | Layer 4 | Layer 8 | Layer 12 | Layer 16 | Layer 20 |
|--------------|---------|---------|----------|----------|----------|
| RouterInterp | **0.60** | **0.64** | **0.61** | **0.62** | **0.60** |
| Bigram Baseline | 0.23 | 0.47 | 0.42 | 0.35 | 0.40 |

Table 3: RouterInterp's natural language explanations predict expert routing with much higher explanation scores than explanations generated from high-frequency token statistics. Explanation scores measure how accurately a language model can predict expert activation given only the explanation. Across different depths of gpt-oss-20b, RouterInterp outperforms the bigram based explanations by 36–161%. Each expert explanation aggregates the 10 most $\rho$-useful SAE features.

## 5.3 Expert Interpretability

To evaluate whether RouterInterp's explanations of routing behaviour are human-understandable, we measure explanation quality using explanation scores (Section 4.4). Table 3 shows results for gpt-oss-20b, where each expert's explanation aggregates explanations of the 10 SAE features with highest $\rho$-usefulness scores for that expert. RouterInterp achieves 0.60–0.64 across layers, substantially outperforming the explanations generated from bigram-expert co-occurrence frequencies (bigram baseline: 0.23–0.47). Beyond this quantitative improvement, there is a qualitative difference: n-gram explanations describe *which tokens* appear frequently, while RouterInterp explanations describe the *concepts* experts respond to. We illustrate this with example explanations in Section F.

## 6 Conclusion

We introduced *RouterInterp*, a method that interprets MoE routing decisions by identifying sparse autoencoder features most predictive of expert selection and aggregating their explanations into unified natural language descriptions. RouterInterp achieves a 0.62 explanation score compared to 0.35 for token-expert co-occurrence methods. RouterInterp emerged from formalizing two alternative specialisation hypotheses: the *Domain Specialisation Hypothesis* (DSH), positing that experts specialize in coherent domains, and the *Superposed Specialisation Hypothesis* (SSH), modeling experts as responding to a superposition of features spanning unrelated micro-domains. Our experiments provide evidence consistent with the SSH: SAE-based routing prediction achieves 81% recall vs. 55% for n-gram baselines; co-activation analysis of features predictive of expert routing shows that such features fire near-independently (median normalised co-activation of 1.00 across 128 experts); and macro-domain analysis finds that experts route tokens from all Pile subsets in near-corpus proportions.

We acknowledge several limitations of our method: (1) RouterInterp is downstream of SAE quality: weak or polysemantic feature explanations propagate to expert-level descriptions. (2) Textual summaries struggle to be both concise and complete as more features are included, so interactive presentations may better serve expert interpretation. We are excited to improve upon these issues in future work.

Several open directions remain. First, the *mechanisms driving expert specialisation* deserve deeper study: why experts converge to particular feature combinations, whether load-balancing or capacity constraints force redundancy, and whether experts cluster by input similarity or functional similarity (i.e. whether co-routed tokens are co-routed because they share semantic content or because they require similar downstream transformations). Second, tracking how routing evolves across *training checkpoints* could reveal when specialisation emerges (Wang et al., 2024; Kangaslahti et al., 2025; Wang et al., 2025).

By demonstrating that SAE features provide both predictive accuracy and semantic interpretability for MoE routing, we take a step toward making expert routing a transparent and controllable component of foundation models. As MoE architectures continue to power frontier systems, tools that reveal the logic behind routing decisions will be essential for ensuring these systems remain aligned with human intent.

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

# A Related Work

## A.1 Sparse Feature Decompositions for Model Interpretability

SAEs rely on the Linear Representation Hypothesis (Mikolov et al., 2013; Bricken et al., 2023; Olah et al., 2020; Nanda et al., 2023; Park et al., 2024): the hypothesis that features in LMs are represented linearly (Bereska & Gavves, 2024). The LRH is often a controversial assumption within interpretability (Smith, 2024; Csordás et al., 2024). However, in our case of interpreting expert routing, routing is usually performed by a linear map and so only linearly accessible information can be important for routing. This makes SAEs uniquely well suited to our problem[10].

SAEs have drawbacks, however. SAEs can struggle to capture logical hierarchical structure (Costa et al., 2025); they do not efficiently capture structures of multi-dimensional features (Engels et al., 2025); they do not generally capture relations across multiple tokens (Lubana et al., 2025); and the structure of Relational Composition (Wattenberg & Viégas, 2024) remains difficult for SAEs.

Most work interpreting representations to date has focused on the impact of intermediate representations on the output logits of a forward pass (Bricken et al., 2023) or on representations in a subsequent layer (Marks et al., 2025). We instead focus on how representations impact the behaviour of a subsequent routing layer.

Inspired by the apparent success of SAE-based explanations, Sharkey (2024) describes a framework for the continual improvement of MI explanations. The framework's three stages are: (1) *mathematical description* (breaking down the neural network into functional parts [11]), (2) *semantic description* (labelling each functional part in a way that is understandable to humans [12]) and (3) *validation* (using the semantic description to make predictions about model behaviour and evaluating these predictions). Here our combination of the trained SAEs and our Ansatz (conjecture) that router weights are linear combinations of weights provide our mathematical description of routing. We leverage AutoInterp (Paulo et al., 2025) to aid with semantic description and evaluation of the functional parts and the empirical success of this approach provides us with evidence for the Ansatz.

## A.2 Mixture of Experts Interpretability

Shazeer et al. (2017b) designed the modern Sparse MoE layer to improve the efficiency and scalability of very large neural networks. Other researchers, however, hoped that the specialisation routing provided might also extend some interpretability benefits (Jacobs et al., 1991; Fedus et al., 2022a). However, Jiang et al. (2024) find it difficult to obtain clean interpretability results with their MoE model Mixtral, remarking "Surprisingly, we do not observe obvious patterns in the assignment of experts based on the topic [of the input text]." Lewis et al. (2021) and Zoph et al. (2022) analyse the unigram patterns of routing and find that the observed level of specialisation varies dramatically making interpretability difficult. Tigges (2025) report some success with unigram analysis of a few experts such as a 'business' expert. We show, however, that by using SAE features as a basis for explanation rather than tokens, we are able to achieve more accurate and concise explanations.

Yang et al. (2025b) suggest that with very large numbers of experts, routing can be made somewhat interpretable. However, their setting requires more experts than is compute optimal given scaling laws for sparse models (Abnar et al., 2025; Krajewski et al., 2024) or more experts than is typical in high performing open source MoE models (Liu et al., 2024; Yang et al., 2025a; Agarwal et al., 2025; Muennighoff et al., 2025).

---

[10]Also note that the Tuned Lens (Belrose et al., 2023) also validates that linear probes on intermediate residual stream states capture meaningful structure.

[11]where here the parts could be in terms of representations (Bricken et al., 2023) or computations (Braun et al., 2025)

[12]note that implicitly this formulation requires the Independent Additivity Principle (Ayonrinde et al., 2024) - if multiple parts are relevant at one time, then it must be the case that we can understand the whole simply from understanding the constituent parts

Chaudhari et al. (2025) provide an alternative model for understanding the phenomena of superposition (Elhage et al., 2022) in MoE models. Chaudhari et al. (2025) find that individual experts exhibit greater monosemanticity than equivalent dense models (in a toy setting). Crucially, their analysis measures monosemanticity at the level of *expert weight matrices*, not at the level of *routing decisions*, which we focus on. We argue that even if individual experts represent their assigned features monosemantically, the routing decision itself may be polysemantic. The router partitions the input space such that features assigned to the same expert only compete with each other for representational capacity (Chaudhari et al., 2025). This creates an incentive to route *dissimilar* domains to the same expert: micro-domains that rarely co-occur can share an expert with little interference, enabling the expert to effectively perform Computation in Superposition (Hänni et al., 2024).

## A.3 ADAPTIVE COMPUTATION

Sparse MoE models are a type of Adaptive Computation model (Graves, 2016; Ayonrinde, 2023b)[13]. Adaptive Computation models are typically either more FLOP or parameter efficient than dense models as they either use a subset of their parameters (for example sparse MoE models or Early Exit models (Banino et al., 2021)) or reuse parameters (for example looping models (Dehghani et al., 2018; Yang et al., 2024)) respectively. There has been little work on the interpretability of Adaptive Computation models and while our work focuses on interpreting routing in the sparse MoE layer, we would be excited about work generalising this approach to routing in early-exit models like Mixture of Depths (Raposo et al., 2024).

## A.4 MODULARITY IN NEURAL NETWORKS

Many brain-inspired neural network architectures employ modularity (the organisation of a system into functional, sparsely connected subunits) as a core component (Pfeiffer et al., 2023), as modularity is believed to be one of the key properties that makes brains efficient and effective (Clune et al., 2013).

We might hypothesise that because neural networks generalise so well and have much lower effective dimensionality than the model dimension would suggest (Lau et al., 2025), we should expect the models to contain intrinsic modularity. That is to say that even though NNs look fully connected and highly entangled, perhaps there is some way of viewing computation such that the computation is in fact highly modularised (Bushnaq et al., 2024). However, efforts to find such modularity have proven difficult (Filan et al., 2021), partially because it is not clear in what form we should expect such modularity to appear.

One way to understand the modularity that neural networks naturally and implicitly form, is to enforce some modular computation and see how the neural networks react. Sparse MoE models have this kind of enforced modularity and we hope that in studying these models we can develop better tools for uncovering possible latent modularity in dense neural networks. In particular, one promising path for understanding modularity may be through the optimisation pressures that resulted in such modularity. For example specialisation and reducing connection costs are two evolutionary pressures for modularity to develop in biological models (Clune et al., 2013; McDougall et al., 2022). Analogously, we might expect that specialisation pressures in the training processes for AI systems also results in the development of modularity as in Wang et al. (2025).

## B EXPERIMENTAL SETUP

**Models** We evaluate on two MoE architectures: OLMoE-1B-7B (Muennighoff et al., 2025) and gpt-oss-20b (Agarwal et al., 2025).

**Datasets.** For experiments with OLMoE, we use OLMoE-mix-0924 (Muennighoff et al., 2025) for SAE training and routing prediction experiments. For gpt-oss, we use a following

---

[13]Also known as Conditional or Dynamic Computation models (Han et al., 2021)

|                          | Layer 4 | Layer 8 | Layer 12 | Layer 16 | Layer 20 |
|--------------------------|---------|---------|----------|----------|----------|
| Unigram Baseline         | 0.570   | 0.602   | 0.523    | 0.573    | 0.553    |
| Bigram Baseline          | 0.631   | 0.639   | 0.560    | 0.593    | 0.589    |
| SAE Predictor ($m = 1$)  | 0.690 | 0.462 | 0.597 | 0.687 | 0.748 |
| SAE Predictor ($m = 4$)  | 0.790   | 0.809 | 0.754 | 0.811 | 0.847 |
| SAE Predictor ($m = 8$)  | 0.803   | 0.824   | **0.811** | 0.800 | 0.844 |
| SAE Predictor ($m = 16$) | 0.817   | 0.830   | 0.810    | **0.830** | 0.861 |
| SAE Predictor ($m = 32$) | **0.824** | **0.865** | 0.801 | 0.820 | 0.866 |
| SAE Predictor ($m = 64$) | 0.822   | 0.858   | 0.793    | 0.825    | **0.873** |

Table 4: Routing prediction improves with more SAE features, consistent with polysemantic expert behaviour predicted by the SSH. Underlined values outperform n-gram baselines; **bold** marks best Recall with fewest features. Notably, even a single feature ($m = 1$) outperforms n-gram baselines at 4 of 5 layers, while performance continues improving up to $m = 32$–$64$, indicating that routing depends on multiple distinct concepts rather than a single domain.

mixture: 75% gpt-oss-generated continuations from The Pile (Gao et al., 2020) prompts[14] and 25% FineWeb (Penedo et al., 2024). We use The Pile (Gao et al., 2020) as a dataset for explanation scoring.

**SAEs.** For OLMoE, we train Top-K SAEs for 100M tokens on activations sampled from layers 3, 7, 11, and 15 (out of OLMoE's 16 layers). Our SAEs have $k = 32$ and 32,768 features (an expansion of 16x compared to the residual stream size). For gpt-oss (24 layers), we use trained BatchTopK SAEs[15] (Lin, 2025) on layers 4, 8, 12, 16, and 20, with 131,072 features and $k = 64$.

**Routing Prediction.** Linear classifiers are trained on SAE features from 1M tokens. We vary the number of active features $m$ to assess how routing information distributes across features. Specifically, for OLMoE we use $m \in \{1, 2, 4, 8, 16, 32\}$, while for gpt-oss we use $m \in \{1, 2, 4, 8, 16, 32, 64\}$.

We compare our SAE-based classifiers to unigram and bigram models that predict routing based on n-gram frequencies in training data. Concretely, for each n-gram (unigram or bigram) in the train set, we collect the number of times it activates each expert. We then predict the activated experts by taking the top-$k$ experts with the highest activation frequency for a given n-gram.

**Expert Interpretation.** For gpt-oss, we generate explanations across layers using the top-10 $\rho$-useful features per expert. We use the Delphi library (Paulo et al., 2025) with Claude Haiku 4.5 (Anthropic, 2025) for explanation generation, summarisation, and calculating explanation scores. We use the all-MiniLM-L6-v2 model from Sentence-Transformers library (Reimers & Gurevych, 2019), for calcuting embeddings for samples in the evaluation set for explanation scoring,

## C  ROUTING DEPENDS ON MULTIPLE FEATURES

In Section 5.1, we reported routing prediction results using $m = 32$ active features. Here we ablate the effect of the number of active features $m$ on routing prediction Recall. Table 4 shows Recall for varying $m \in \{1, 2, 4, 8, 16, 32, 64\}$ across five layers of gpt-oss-20b.

The results reveal two key patterns. First, Recall continues to improve with more features across all layers, indicating that routing decisions depend on multiple distinct concepts —consistent with polysemantic expert behaviour. Second, even a single feature ($m = 1$) outperforms n-gram baselines at most layers (underlined), demonstrating that individual SAE features carry substantial routing information. This aligns with our finding that $\rho$-useful features achieve strong explanation performance with just 1–2 features (Appendix D).

---

[14]https://huggingface.co/datasets/andyrdt/gpt-oss-20b-rollouts
[15]https://huggingface.co/andyrdt/saes-gpt-oss-20b

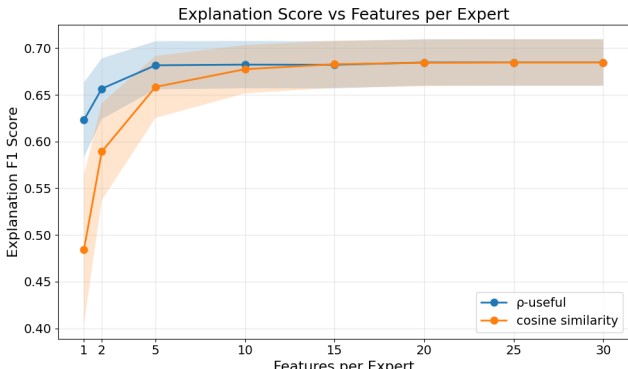

Figure 3: The most useful features (ranked by the $\rho$-usefulness metric) are individually *more* predictive of expert routing than features with highest cosine similarity with router weights. Despite selecting largely non-overlapping feature sets, both methods converge to similar performance at 10 features, suggesting routing information is redundantly encoded across many SAE features. Results are reported for layer 16 of gpt-oss-20b. Shaded regions show standard deviation across experts.

## D    FEATURE SELECTION METHOD AND SET SIZE

In Section 4.2, we describe selecting the top-$n$ features by $\rho$-usefulness for each expert. Here we justify this choice by evaluating (1) how the feature set size $n$ affects explanation quality, and (2) how $\rho$-usefulness compares to an alternative selection method based on cosine similarity.

As an alternative to $\rho$-usefulness, we consider selecting features based on the geometric alignment between SAE decoder directions $\boldsymbol{d}_f \in W_{dec}$ and router weight vectors $\boldsymbol{w}_k \in \boldsymbol{W}_r$. For each expert $E_k$, we rank features $f$ by $\cos(\boldsymbol{d}_f, \boldsymbol{w}_k)$ and select the top-$n$.

To isolate the effect of feature selection, we replace the LLM summarisation with a simpler aggregation: we evaluate each feature's explanation independently and label an example as positive if *at least one* feature's explanation predicts expert activation. This setup directly measures whether adding more features increases predictive coverage: as the feature set grows, we track whether additional features contribute new predictive information.

Figure 3 shows that individual $\rho$-useful features carry strong predictive signal— even a single feature achieves F1 $\approx 0.62$. In contrast, features with high cosine similarity to router weights are only predictive in aggregate (F1 $\approx 0.49$ at $n = 1$), suggesting that geometric alignment with router weights does not guarantee a feature fires when that expert is selected. Both methods converge to approximately the same explanation score ($\approx 0.68$) at 10 features.

The convergence at 10 features is interesting given that the two criteria select largely non-overlapping feature sets. This suggests that routing-relevant information is encoded across many SAE features, and with enough features, both methods eventually capture sufficient signal to predict routing.

## E    EVIDENCE FOR SUPERPOSED SPECIALISATION

This appendix provides extended results for the two experiments summarised in Section 5.2, which test whether expert routing is better explained by the Superposed Specialisation Hypothesis (SSH) or the Domain Specialisation Hypothesis (DSH).

### E.1    FEATURE CO-ACTIVATION ANALYSIS

For each expert $E_i$ in gpt-oss-20b, we select the top-20 $\rho$-useful SAE features and measure pairwise co-activation rates across all $\binom{20}{2} = 190$ feature pairs, evaluated over $\sim$10M tokens. A feature pair co-activates when both features fire (activate above zero) on the same token. We compute the mean co-activation rate $P(f_j$ fires $\wedge\ f_k$ fires$)$ for each expert's feature set.

To assess whether features are related or independent, we normalise the observed co-activation by the rate expected under statistical independence: $P(f_j) \cdot P(f_k) \approx \bar{p}^2$, where $\bar{p}$ is the mean firing rate of the expert's features. A normalised ratio of 1.0 indicates that features fire independently of each other; values substantially above 1.0 would indicate that features tend to co-occur (as expected under the DSH, where features represent the same domain); values near or below 1.0 indicate unrelated activation patterns (as expected under the SSH).

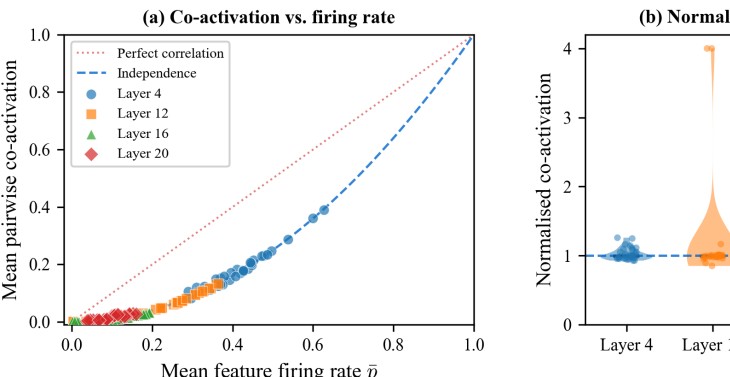

Figure 4: Expert-predictive features co-activate at rates consistent with statistical independence, supporting the SSH over the DSH. **(a)** Each point represents one expert in gpt-oss-20b; observed co-activation follows the independence curve $\bar{p}^2$ (dashed blue) rather than the perfect-correlation line $\bar{p}$ (dotted red). **(b)** The normalised co-activation ratio clusters around 1.0 across all layers. Across 128 experts (32 per layer), 88% have normalised ratios in the range $[0.5, 2.0]$, and the overall median is 1.00. This indicates that expert-predictive features represent semantically disjoint micro-domains rather than a single coherent domain.

Figure 4 presents the results across layers 4, 12, 16, and 20. In panel (a), observed co-activation rates closely track the independence curve $\bar{p}^2$ rather than the perfect-correlation line $\bar{p}$. Panel (b) shows the distribution of normalised co-activation ratios: the median ratio is 1.00 across all 128 experts, and 88% of experts (113/128) fall within the $[0.5, 2.0]$ range around independence. At layer 4, where features have the highest firing rates ($\bar{p} \approx 0.42$), all 32 experts exhibit near-independent co-activation (median ratio 1.00). At layer 16, the median ratio drops to 0.82, suggesting mildly anti-correlated features. At layer 20, the median rises to 1.48, indicating slight positive correlation, though still well below what would be expected under the DSH. Additionally, 44% of experts contain at least one feature pair that never co-activates across the entire evaluation set, indicating completely disjoint activation patterns.

These results demonstrate that the features most predictive of each expert's routing fire largely independently of one another. This is consistent with the SSH—each expert responds to a collection of unrelated micro-domains—and inconsistent with the DSH, which would predict strong positive co-activation among features within the same expert.

### E.2    THE PILE CASE STUDY

We complement the micro-domain analysis above with a macro-domain analysis. If the DSH holds, experts should preferentially route tokens from particular corpus domains; if the SSH holds, each expert should draw tokens from a broad mix of domains. (Jiang et al., 2024) performed a similar analysis for Mixtral, plotting the proportion of tokens from different domains assigned to each expert, but found no obvious topic-based patterns. We follow their approach using the Pile dataset (Gao et al., 2020), which comprises diverse subsets—including ArXiv, PubMed, GitHub, StackExchange, Wikipedia, and FreeLaw, among others—spanning scientific, legal, code, and web domains. We process ~10M tokens over $N_{\text{subsets}} = 16$ Pile subsets. For each expert $E_i$, we compute the proportion of tokens from each subset that are routed to $E_i$, yielding a per-expert distribution over subsets.

We quantify domain specialisation via **normalised entropy**: the Shannon entropy (Shannon, 1948) of each expert's subset distribution divided by $\log N_{\text{subsets}}$, so that 1.0 corresponds to a perfectly uniform distribution over all subsets (no specialisation) and 0.0 corresponds to routing from a single subset only (maximum specialisation).

**Corpus-proportional baseline.** The Pile subsets are not equally sized, so even an expert with no domain preference will not have an entropy of exactly 1.0. The appropriate reference baseline is the *corpus-proportional* distribution: the entropy an expert would exhibit if it mirrored the global mix of Pile subsets exactly, i.e. if its per-subset routing shares were proportional to the corpus-wide token counts. This baseline evaluates to 0.8 (normalised); any expert whose entropy matches this value is simply reflecting the composition of the corpus rather than specialising.

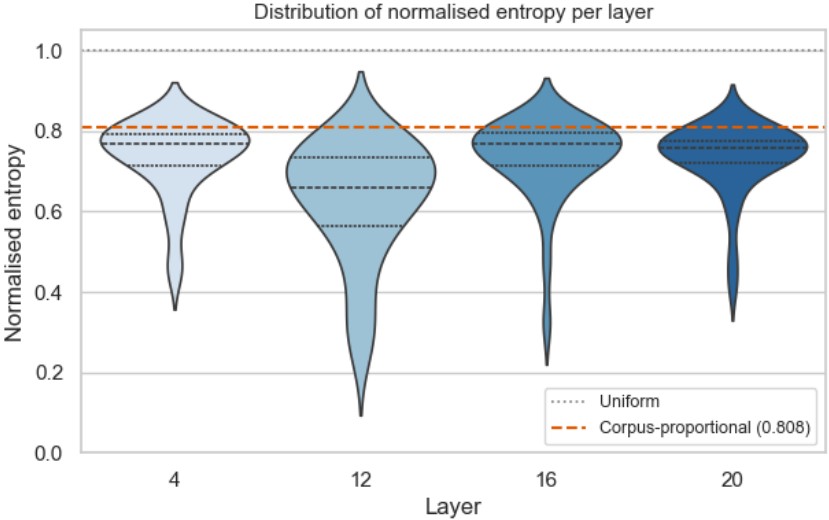

Figure 5: Experts show little macro-domain specialisation across 16 Pile subsets. Each violin shows the distribution of normalised entropy of each expert's Pile subset routing distribution across 32 experts at a given layer of gpt-oss-20b. The dashed line marks the corpus-proportional baseline (0.8): the entropy an expert would have if it simply mirrored the unequal sizes of the 16 Pile subsets. Expert entropies cluster near this baseline at every layer, indicating that routing reflects corpus composition rather than domain preference.

As shown in Figure 5, expert entropies cluster tightly around the corpus-proportional baseline across all layers, with very few experts deviating substantially. This confirms that experts route tokens from all Pile subsets in approximately the same proportions as the corpus overall.

Taken together with the co-activation analysis in Section E.1, both experiments point in the same direction: the majority of experts do not specialise in a single coherent domain, at either the feature level or the dataset level, consistent with the SSH. However, neither experiment is without outliers. A handful of experts exhibit noticeably lower entropy in Figure 5 (particularly at layer 12), and a small fraction show elevated co-activation ratios in Figure 4. These outliers suggest that while the SSH captures the dominant pattern, a minority of experts may exhibit partial domain alignment.

## F   EXAMPLE EXPERT EXPLANATIONS

Table 5 shows different levels of RouterInterp explanations for a randomly selected expert (Expert 8) from layer 16 of gpt-oss-20b. It illustrates how expert-level explanations aggregate multiple feature-level explanations, which are in turn acquired by explaining examples where the feature activates.

| Level | Explanation |
|---|---|
| **Expert 8** | This expert activates across five distinct contexts: (1) Structural and grammatical elements—function words, articles, prepositions, conjunctions, and punctuation that organize text and parse grammatical relationships. (2) Informational and technical content—medical/clinical terminology, dates, measurements, quantitative data, multilingual technical terms, and metadata markers, with stronger activation in content-dense, semantically specific text. (3) Logical and reasoning anchors—punctuation marking boundaries, conjunctions signaling relationships, mathematical operators, and discourse markers in technical or argumentative contexts. (4) Self-identification phrase—high activation on tokens within "a large language model trained by OpenAI" in system messages. (5) Mathematical and technical notation—numerical values, operators, variables, LaTeX notation, and domain-specific keywords encoding mathematical constraints and relationships. |
| *Feature 46328* | *Self-identification phrase—tokens within "a large language model trained by OpenAI" in system messages, particularly " large," " model," " trained," and "AI."* **Activating Examples:** [1] Chat GPT , a large language model trained by Open AI . |
| *Feature 70455* | *Logical and reasoning anchors—punctuation marking boundaries, conjunctions signaling relationships, mathematical operators, and discourse markers in technical or argumentative contexts.* **Activating Examples:** [1] a GUI ( like Chess Base , Arena , Stockfish 's own GUI , or |
| *Feature 73639* | *Structural and grammatical elements—articles, prepositions, conjunctions, punctuation, and mathematical delimiters that organize and parse text.* **Activating Examples:** [1] and $ y $ . Let $ n $ be the number of possible values of [2] s 393 56 ash Maybe it's best to ignore |
| *Feature 81948* | *Informational and technical content—medical/clinical terms, dates, measurements, quantitative data, multilingual technical terms, and metadata markers. Higher activation in content-dense, semantically specific text.* **Activating Examples:** [1] though he termed it " spec ulative ." He acknowledged [2] I 've never been through anything like this before . |
| *Feature 91358* | *Mathematical and technical notation—numerical values, operators, variables, LaTeX notation, and domain-specific keywords encoding mathematical constraints and relationships.* **Activating Examples:** [1] But it's given that he secured 110 marks . So 170 - T [2] 2 - y _ 1 } + \{} frac { y _ 2 ^ 2 }{ |

Table 5: RouterInterp first explains features by looking at examples where these features activate strongly, and then aggregates these explanations into an expert-level explanation. For example, Expert 8 from gpt-oss-20b layer 16 specialises in AI self-identification (*Feature 46328*), medical terms (*Feature 81948*), mathematical notation (*Feature 91358* and *Feature 73639*), and reasoning anchors (*Feature 70455*). Here we show explanations for the top 5 $\rho$-useful SAE features (rather than 10) for brevity.

