# OpenReview forum: "RouterInterp: Understanding Superposed Specialisation in MoE Routing"
_ICLR.cc/2026/Workshop/Sci4DL — Sci4DL 2026_

### Official Review · Reviewer_VGcg · 2026-02-10

**Fit:** 2
**Significance:** 2
**Confidence:** 2

**Summary:**

This paper investigates the routing mechanism in Sparse Mixture of Experts (MoE) models by formalizing the Superposed Specialisation Hypothesis (SSH). Mathematically, it rejects the Domain Specialisation Hypothesis (DSH), which posits that experts $E_i$ represent a single semantic domain $d \in \mathcal{D}$. Instead, SSH models each expert as a disjoint union of fine-grained features extracted via Sparse Autoencoders (SAEs). The authors introduce RouterInterp, identifying the subset of SAE features $\mathbb{F}_i \subset \mathcal{F}$ that maximize the p-usefulness metric $\rho(f, E_i)$. The paper provides evidence that routing is better explained by these linear feature representations than by surface-level token statistics, achieving a Recall of $0.81$–$0.89$ in predicting expert activation.

**Strengths:**

1. P-Usefulness Formulation: The introduction of the p-usefulness metric $\rho(f, E_i) = \mathbb{E}[f(x)|E_i \in \mathcal{T}(x)] - \mathbb{E}[f(x)|E_i \notin \mathcal{T}(x)]$ provides a rigorous, data-driven criterion for feature selection. It specifically measures the discriminative power of a feature for a routing decision rather than mere correlation.

2. Validation of the Linear Representation Hypothesis (LRH): By training a linear classifier on sparse representations $z$, the authors demonstrate that routing relies on the LRH. This justifies the use of SAEs, as MoE routing is typically a linear map $h(x) = W_r \cdot x$, meaning only linearly accessible information can influence the gate.

3. Empirical Robustness Across Depth: The paper provides a layer-wise analysis (Layers 4–20) showing that the Recall for SAE-based predictors remains consistently high ($0.79$–$0.85$), while token-based baselines plateau. This suggests that routing complexity is an invariant property of the model's depth.

4. Formalization of Load-Balancing Constraints: The authors provide a theoretical argument that the load-balancing loss $\mathcal{L}_{aux}$ encourages SSH by forcing tokens from the same macro-domain to be distributed across experts, essentially necessitating polysemantic behavior to satisfy uniform utilization.

**Suggestions:**

Critical Weaknesses and Questions for the Authors

1.
Correlation isn't Causality (SSH Proof): The paper shows that certain features can *predict* when an expert turns on, but it doesn't prove the expert is actually *using* those features to do its job. Just because a feature and an expert light up at the same time doesn't mean the expert is performing "Computation in Superposition".


* **Question**: Have you tried any "causal" tests, like **activation steering** or **path patching**? If you kill off one of these superposed features, does the expert’s performance on that specific task drop while its other tasks stay healthy?


2.
The ρ-usefulness Statistic is a Bit "Raw": Using a simple mean difference for p-usefulness is risky because it doesn't account for how much those activations swing. Features that are just "noisy" or appear in boilerplate text (like system prompts) could easily game this metric without being semantically important.


* **Question**: Why not use a normalized metric like a **t-statistic**, **Cohen’s d**, or **Mutual Information**? This would ensure you aren't just picking up on high-variance noise or dataset quirks.


3.
The Baselines Feel Too Easy: Comparing a sophisticated Sparse Autoencoder (SAE) to simple word-counting (n-grams) isn't exactly a fair fight. We need to know if this "superposition" is actually happening inside the experts, or if it’s just a property of the data itself.


* **Question**: How does your SAE predictor compare to a **linear probe** run directly on the residual stream? If a simple probe works just as well without the SAE, then this "superposition" might not be an expert-level phenomenon at all.


4.
 Grading Your Own Homework (LLM Bias): You're using an LLM to explain the features and then using another LLM to score those explanations. Since these models often share the same training data and "logic," they might just be agreeing with each other's linguistic patterns rather than the actual model weights.


* **Question**: Did you try **cross-model evaluation**? For example, having one model type generate the explanation and a completely different architecture (or a human) score it to make sure the results aren't just a byproduct of shared LLM "vibes."


5.
Are the Domains Truly "Disjoint"?: You argue that experts handle unrelated "micro-domains," but you haven't mathematically proven they are actually unrelated. A "Physics" expert using features for "Math" and "LaTeX" isn't polysemantic- those things belong together.


* **Question**: Do you have a quantitative way to show these features are **semantically orthogonal**? We need to see that an expert is handling truly "random" combinations (like "French law" and "Python code") to believe the SSH over the DSH.


6.
Is Load-Balancing the Real Culprit?: The paper suggests that forcing the model to use all experts (load balancing) is what causes this messy superposition. It's a great theory, but we don't see the "before and after".


* **Question**: Have you looked at MoE models trained **without** a load-balancing loss? Seeing if those experts stay "cleaner" and more domain-specific would go a long way in proving your theory about training pressures.

---

### Official Review · Reviewer_nDA6 · 2026-02-15

**Fit:** 3
**Significance:** 2
**Confidence:** 2

**Summary:**

In this paper, authors challenge the common assumption that experts in Sparse Mixture-of-Experts (MoE) models specialize in single, coherent semantic domains. Instead, it proposes the Superposed Specialisation Hypothesis (SSH), under which each expert responds to a disjoint set of fine-grained features of multiple unrelated micro-domains.

To show this, the authors introduce RouterInterp, a method that interprets MoE routing decisions using sparse autoencoder (SAE) features. It identifies the features that are most predictive of expert selection, generates natural language explanations for them, and aggregates them into expert-level explanations. Their experiments show that this explanations substantially outperform token-based baselines in predicting expert routing, providing empirical support for SSH and a scalable approach to MoE interpretability.

**Strengths:**

- **Clear problem framing and motivation**: The paper clearly presents the current interpretability gap in MoE models and formalizes two competing hypotheses (DSH vs SSH). They motivate with prior negative results and use intuitive figures for support.

- **Methodological clarity**: RouterInterp is described in a structured and easy-to-follow way, with clear separation and description of feature selection, feature explanation, and aggregation, as well as how to evaluate the quality of the method.

- **Scalability and relevance**: The method is demonstrated on large, realistic MoE models rather than toy settings, which strengthens the relevance of the results for practical interpretability work on state-of-the-art frontier models.

- **Limitations**: method limitations are acknowledged and some lines and ideas for future work are presented.

**Suggestions:**

- **Limited presentation of qualitative results in the main paper**: While the appendix includes more detailed results and baselines, as well as example explanations, the main body contains relatively few qualitative results to support RouterInterp.

- **Sensitivity to SAE quality and configuration**: RouterInterp seems to depend heavily on the quality and hyperparameters of the underlying sparse autoencoders (feature count, sparsity level, training data...). While this dependency is acknowledged, its practical impact (e.f. through relevant ablations) could be further explored.

- **Scope of the SSH**: The evidence supports SSH for the examined architectures and routing layers, but it is not fully clear how broadly the hypothesis generalizes to different routing mechanisms or training regimes.

- (Minor) Style files seem to have not been respected, the font used is different than the one provided in the template.

---

### Official Review · Reviewer_sdde · 2026-02-23

**Fit:** 2
**Significance:** 2
**Confidence:** 2

**Summary:**

Generally, the hypothesis that Mixture of Experts models have highly specialized/coherent experts has not been found to be true. As an alternative to this hypothesis, the authors suggest that experts may represent disjoint concepts--perhaps driven by this representation structure allowing higher representational capacity (multiple experts can use different transformations on similar topics) and load balancing auxiliary loss. To test this, authors find the most meaningful SAE features for a given expert and use those features to create explanations (and predictions) for each expert. This is more effective than doing predictions directly using token-level information.

**Strengths:**

* The presentation is quite clear, all of the techniques are explained well
* There's not much work on directly interpreting MoE experts nor a clear hypothesis as to what inductive biases we should incorporate into our methods--this work helps provide that
* The authors acknowledge potential weaknesses of their method

**Suggestions:**

* I would've liked something exploring just how disjoint the predictive features are. What is the coactivation rate of the predictive feature set for an expert vs a randomly selected group of the same size? This would directly support the SSH hypothesis. Otherwise, I worry the metrics presented can be tautological--if the SAE reconstructs the residual stream well and every expert doesn't activate on every input, then I'd expect there to be some subset of SAE features that predict expert activation well. The diminishing returns of larger sets of features is at least suggestive however that there is a fairly specifiable/distinct set of concepts which each expert has
* As is standard for this kind of work, the strength of the autointerp pipeline presents difficulties. For instance, the expert explanation example isn't super compelling. Including some activating examples for the expert itself could help

---

### Meta-Review · Area_Chair_pSaU · 2026-03-01

**Recommendation:** Accept

**Metareview:**

Recommending acceptance, a decent fit for the workshop and interesting work.

---

### Decision · Program_Chairs · 2026-03-02

Accept